# Dehydration Status Aggravates Early Renal Impairment in Children: A Longitudinal Study

**DOI:** 10.3390/nu14020335

**Published:** 2022-01-13

**Authors:** Nubiya Amaerjiang, Menglong Li, Huidi Xiao, Jiawulan Zunong, Ziang Li, Dayong Huang, Sten H. Vermund, Rafael Pérez-Escamilla, Xiaofeng Jiang, Yifei Hu

**Affiliations:** 1Department of Child, Adolescent Health and Maternal Care, School of Public Health, Capital Medical University, Beijing 100069, China; 13693617970@163.com (N.A.); limenglong_ph@163.com (M.L.); xhd19988023@163.com (H.X.); jiawulan@foxmail.com (J.Z.); 15622764530@163.com (Z.L.); vincejiang@163.com (X.J.); 2Department of Hematology, Beijing Friendship Hospital, Capital Medical University, Beijing 100050, China; hdayong@yahoo.com; 3Yale School of Public Health, Yale University, New Haven, CT 06510, USA; sten.vermund@yale.edu (S.H.V.); rafael.perez-escamilla@yale.edu (R.P.-E.)

**Keywords:** dehydration, renal impairment, children, weekly activity patterns

## Abstract

Dehydration is common in children for physiological and behavioral reasons. The objective of this study was to assess changes in hydration status and renal impairment across school weekdays. We conducted a longitudinal study of three repeated measures of urinalysis within one week in November 2019 in a child cohort in Beijing, China. We measured urine specific gravity (USG) to determine the dehydration status, and the concentration of β_2_-microglobulin (β_2_-MG) and microalbumin (MA) to assess renal function impairment among 1885 children with a mean age of 7.7 years old. The prevalence of dehydration was 61.9%, which was significantly higher in boys (64.3%). Using chi-square tests and linear mixed-effects regression models, we documented the trends of the renal indicators’ change over time among different hydration statuses. Compared to Mondays, there were apparent increases of β_2_-MG concentrations on Wednesdays (β = 0.029, *p* < 0.001) and Fridays (β = 0.035, *p* < 0.001) in the dehydrated group, but not in the euhydrated group. As for the MA concentrations, only the decrease on Fridays (β = −1.822, *p* = 0.01) was significant in the euhydrated group. An increased trend of elevated β_2_-MG concentration was shown in both the euhydrated group (Z = −3.33, *p* < 0.001) and the dehydrated group (Z = −8.82, *p* < 0.001). By contrast, there was a decreased trend of elevated MA concentrations in the euhydrated group (Z = 3.59, *p* < 0.001) but not in the dehydrated group. A new indicator ratio, β_2_-MG/MA, validated the consistent trends of renal function impairment in children with dehydration. Renal impairment trends worsened as a function of school days during the week and the dehydration status aggravated renal impairment during childhood across school weekdays, especially tubular abnormalities in children.

## 1. Introduction

Water is an important component of the human body and plays a vital role in maintaining human life and metabolic health [1]. Inadequate fluid intake can leave the body in a dehydration status, especially in children, and dehydration due to physiological and behavioral reasons may have an impact on physical activity [2], cardiovascular health [3], cognitive performance [4], and renal function [5].

The greatest fluid requirements relative to body weight are extant during childhood, and the larger body surface area, and consequently the higher insensible water loss through the skin, puts children at greater risk for dehydration than adults [6]. Fluid drinking patterns and high activity levels during school play, sports, or performances play critical roles in hydration [7]. Previous studies have shown that elementary school children often start the school day in a status of mild dehydration [8]. In China, high academic pressures [9] often lead to students to having short inter-curriculum rest periods; children may not drink enough fluids in order to reduce the micturition frequency. Weekly activity patterns from Monday to Friday include periodic insufficient water intake and long periods of sedentary behavior. Both can have a negative impact on children’s health, particularly renal damage.

The concentrations of β_2_-microglobulin (β_2_-MG) and microalbumin (MA) in the urine are markers of early renal impairment [10,11]. Urine β_2_-MG helps assess the damage or dysfunction on the tubules [12] and MA reflects the early renal injury of glomerular [13]. Studies on urine β_2_-MG and MA in children have focused on kidney-related diseases [14], diabetes [15], urinary tract infections [16], blood pressure [17], and the prevalence of early renal impairment among children and adolescents [18,19,20]. To our knowledge, there are no longitudinal studies investigating the effect of hydration status on early renal impairment in school children relating to weekly activity patterns.

In the present study, we sought to investigate the trends of β_2_-MG and MA at different hydration statuses during a week, and to assess the effect of the hydration status on early childhood renal impairment. We hypothesize that children in a dehydration status may aggravate early renal impairment.

## 2. Materials and Methods

### 2.1. Study Design and Participants

This study is based on sequential baseline surveys of the PROC study in the urban area of Shunyi district, Beijing, which was launched in October, 2018, among 1914 children aged 6–8 years in six non-boarding primary schools (detailed elsewhere [21]). The study was registered at China Clinical Trial Registration (www.chictr.org.cn/enIndex.aspx, no. ChiCTR2100044027, registration on 6 March 2021).

We conducted three repeat measures of urinalysis (UA) in two weeks of November 2019. The urine samples were consecutively collected in the morning of three weekdays (Monday, Wednesday, and Friday) within one week to comprehensively explore the renal damage trends during school time each week: Monday reflected the renal status before the start of the school week after fully “resting” over the weekend, Wednesday reflected the intermediate time, and Friday reflected the accumulation of almost whole week of school time.

Parents helped collect the sample to a sterilized container distributed by the research team and their children transported the specimens to us with the assistance of the teachers. We included PROC participants who completed at least one of the three urine collections. A total of 1855 children (96.9% of the cohort) were included for the present study. Figure 1 shows the participants’ recruitment and selection process.

### 2.2. Anthropometric Measurements

Anthropometric measurements were performed by trained staff from October to November 2018 (detailed elsewhere [21]). Briefly, we measured standing height, rounded to 0.1 cm, averaging two measurements. We measured weight of participants in light clothes using a body composition analyzer (Seehigher H-Key 350, Shanxi, China) and rounded to 0.1 kg. Body mass index (BMI) was calculated as weight in kilograms divided by height in meters squared (kg/m^2^). Waist circumference (WC) was measured twice to 0.1 cm using inelastic tape, validated by steel ruler, at the midpoint horizontal level that links the iliac crest and the lower margin of the 12th rib. The result was the average of the two measurements. *Z*-scores for age and sex of height, weight, and BMI were calculated per 2007 World Health Organization standards [22].

### 2.3. Blood Pressure

In June 2019, blood pressure was measured three times a day using an electronic sphygmomanometer (OMRON HBP-1300, Dalian, China). Systolic blood pressure (SBP) and diastolic blood pressure (DBP) were averaged from the last two measurements.

### 2.4. Blood Collection and Laboratory Assay

Three fasting blood samples (2 mL and 5 mL anticoagulant, 5 mL procoagulant) were collected from a forearm vein in the morning. The serum was separated from the 5 mL procoagulant sample after centrifugation (1000× *g*, 10 min), stored, and transferred immediately with the 2 mL anticoagulant sample to the local hospital for testing. 

Serum lipid panels were measured using automatic clinical chemistry analyzer Beckman Coulter AU5800 (Shizuoka, Japan), including blood glucose (BGLU), total cholesterol (TC), triglyceride (TG), high-density lipoprotein cholesterol (HDL-C), and low-density lipoprotein cholesterol (LDL-C).Hemoglobin A1c (HbA1c) was measured using the automated glycohemoglobin analyzer Tosoh’s HLC-723G8 (Yamaguchi, Japan).

### 2.5. Repeat Measurements of Urine Assay

Repeat urine measurements were completed on Monday, Wednesday, and Friday. For Monday collections, we collected 24-h urine from Sunday morning to the following Monday morning. Fasting morning urines were collected on Wednesdays and Fridays. The β_2_-microglobulin (β_2_-MG) and microalbumin (MA) were measured using the automatic clinical chemistry analyzer Beckman Coulter AU680 (Osaka, Japan). We generated a new indicator ((β_2_-MG/ MA) × 10^3^) to synthesize trends for these two indicators. 

The abnormal value cutoffs were >0.2 mg/L for β_2_-MG [14], ≥20 mg/L for MA [23], and ≥40 for β_2_-MG/ MA.

### 2.6. Hydration Status

Fasting morning urines on Wednesday or Friday were used to measure urine specific gravity (USG) to assess hydration status. Due to many samples needing to be studied in a short time frame, USG was measured by two laboratories (both of them were audited and certified by the National Medical Products Administration) and three instruments: one laboratory used the URIT-500B (Guangxi, China), another used the Dirui H-800 (Guangdong, China) and Combi Scan 500 (Lichtenfels, Germany). Based on USG, the participants were divided into two groups: the USG < 1.020 for the euhydrated group (707 participants) and USG ≥ 1.020 for the dehydrated group (1148 participants) [24,25].

### 2.7. Ethical Consideration

The protocol of the child cohort designed to study sensitization, puberty, obesity, and cardiovascular risk (PROC) was reviewed and approved by the Ethics Committee of Capital Medical University (No. 2018SY82) that applied the guidance of the Declaration of Helsinki, later amendments, and comparable international ethical standards.

### 2.8. Statistical Analysis

The main outcome indicators were the concentrations of β_2_-MG, MA, and ((β_2_-MG/ MA) × 10^3^) over the school week. Descriptive statistics were presented using counts and percentages to describe the categorical variables, and the mean ± standard deviation (SD) and median and interquartile range to describe continuous variables. Except age and sex, the variables having missing values were estimated using multiple imputation to generate a complete dataset. The final analyses were based on 37 imputed datasets. We performed independent *t*-tests, Mann–Whitney U tests, and χ^2^ tests to compare the difference between the euhydrated and dehydrated groups. After standardizing the data for different covariates, we analyzed the temporal trends of β_2_-MG, MA, and ((β_2_-MG/ MA) × 10^3^) using linear mixed-effects regression models with fixed effects using PROC MIXED. The results are shown in Table 1, Figure 2, Table 2 and Table 3. We generated valid statistical inferences for the parameters (Table 2) based on 37 datasets using PROC MIANALYZE, and other results are based on the first imputed datasets. Model 1 analyzed the trend of the individual indicator for one week. Model 2, Model 3, and Model 4 presented the effects after adjusting covariates. A two-sided *p* < 0.05 was the criterion for the statistical significance. All data were analyzed using Statistical Analysis System V.9.4 (SAS Institute Inc., Cary, NC, USA).

## 3. Results

### 3.1. Sociodemographic Characteristics

A total of 1855 children aged 6.7 ± 0.30 years old upon cohort launching were enrolled in this study in 2018. They were measured for blood pressure repeatedly three times in June 2019 and repeated urine sample collection in November of 2019. The prevalence of dehydration was 61.9% (1148 of 1855), which was significantly higher in 927 boys (64.3%). The mean age of urine collection was 7.65 years old. Children with dehydration had a higher Z-score of weight and Z-score of BMI, WC, SBP, and TG compared to children with euhydration. The three measures of the MA level of children with dehydration were significantly higher than that of the children with euhydration, while the differences in the β_2_-MG of Monday’s measure was not significant (Table 1).

### 3.2. Three Urine Indicators’ Percentile Change from Monday to Friday

From Monday to Wednesday to Friday, the average concentration of β_2_-MG (median line) increases and then the trends level, and the p5 line remains stable among both the euhydrated and dehydrated groups, while the p95 line goes upward and then downward in children with euhydration and trends upward in children with dehydration (Figure 2a). The average concentration of MA (median line) remains level and then trends downward in children with euhydration and fluctuates in children with dehydration, and the p5 line remains stable, while the p95 line trends downward in both groups (Figure 2b). For β_2_-MG/ MA, the median line and the p95 line trend upward continuously, while the p5 line trends upward and then levels in both groups (Figure 2c).

### 3.3. Temporal Trends of Renal Damage Indicators (Continuous Value) by Hydration Status

A linear mixed-effects regression model presents the temporal change of the renal damage indicators over time and shows the differences between the euhydration and the dehydration status. There was no statistically significant increase of β_2_-MG in the euhydrated group. Compared to Monday, there were apparent increases in the dehydrated group on Wednesday (β = 0.029, *p* < 0.001) and Friday (β = 0.035, *p* < 0.001). For MA, the results show a decreasing trend, but only the decrease on Friday (β = −1.822, *p* = 0.01) in the euhydration group was statistically significant compared with Monday. As for the ratio of β_2_-MG/MA, an obvious increasing trend was present in both groups compared with Monday: Friday (β = 1.963, *p* = 0.005) in the euhydrated group, and Wednesday (β = 1.687, *p* = 0.003) and Friday (β = 3.559, *p* < 0.001) in the dehydrated group. The increase on Wednesday (β = 1.928, *p* = 0.14) in the euhydrated group was not significant. After adjusting for different covariates, the results remain consistent in the different models (Table 2).

### 3.4. Temporal Trends in the Percentage Change of Renal Damage (Categorical Value) by Hydration Status

Elevated β_2_-MG concentrations increase during the week in both the euhydrated group (Z = −3.33, *p* < 0.001) and the dehydrated group (Z = −8.82, *p* < 0.001) from Monday to Friday. The elevated β_2_-MG concentrations on Wednesday (28.5%) and Friday (32.5%) in the dehydrated group were significantly higher than that (21.1%, 22.8%) of the euhydrated group. The elevated MA concentrations trended downward in both the euhydrated group (Z = 3.59, *p* < 0.001) and the dehydrated group (Z = 0.28, *p* = 0.78). The elevated MA concentrations on Wednesday (5.2%) and Friday (5.3%) in the dehydrated group were significantly higher than that (2.7%, 1.8%) of the euhydrated group. The ratio of β_2_-MG/MA presented an elevated trend in both the euhydrated group (Z = −2.23, *p* = 0.026) and the dehydrated group (Z = −2.72, *p* = 0.007), while the prevalence of elevated ratios of β_2_-MG/MA in the euhydrated group were higher than that in the dehydrated group, though this may have been due to chance (Table 3).

## 4. Discussion

This is the first longitudinal study to assess the relationship between hydration status and renal impairment in elementary school children through their school week. We found that children in different hydration statuses had different temporal trends and varied elevations in β_2_-MG and MA. Those children with dehydration can experience aggravated early renal impairment.

Our finding shows the prevalence of dehydration (determined by USG) was 61.9% (64.3% in boys) in children aged 7.7 years, i.e., grade two of primary school, which is similar to previous studies reporting that over half of children in the U.S. [26] and two-thirds of children and adolescents in China [25] were in a dehydration status at the time of the studies. A cross-sectional study [27] conducted among 18–23 years old college students in China showed that 23.3% of boys were in a dehydration status (determined by urine osmolality), and the significant sex difference is consistent with our study. Sex difference patterns in dehydration status may begin in childhood and persist into adulthood. The prevalence discrepancy may be due to the classification of the hydration status and the physiological and behavioral reasons such as immaturity and restrictions of water and restroom access. Many elementary school students were in a state of dehydration [8]. Why elementary school children are often dehydrated may be related to children’s ability to perceive thirst [28], teachers’ attitudes [29], and the environment of school water facilities and toilets [30]. Many children may choose to reduce water intake to reduce micturition frequency during short inter-curricular breaks.

We observed differences in the Z-score of body weight, Z-score of BMI, and WC between the euhydrated and the dehydrated group. These finding validates a previous study that children with excess body fat have a higher risk of dehydration [31]. Children with dehydration have higher BMI Z-scores and have higher water requirements because water requirements are related to a higher metabolic rate, body surface area, and body weight. Furthermore, obese children are more likely to consume foods which are higher in calories, yet lower in water [32]. Moreover, we observed that children with dehydration were more likely to have higher SBP, TG, and LDL-C and lower HbA1c (the latter two were of borderline significance) because dehydration may increase angiotensin II and blood viscosity and may impair the endothelial function [33].

We found that children with a dehydration status had higher β_2_-MG and MA than those in a euhydration status. β_2_-MG is a low molecular weight protein that is filtered by the glomerulus and completely reabsorbed and catabolized in the renal tubules [34]. Under normal physiological conditions, only a small amount of β_2_-MG can be detected in the urine, and the elevated concentration of urine implies proximal tubular dysfunction [35]. Among children, tubular dysfunction can be used to diagnose tubulointerstitial diseases and the localization of urinary tract infections. Furthermore, β_2_-MG has been reported as a candidate biomarker for acute kidney injury (AKI) [36]. We found an increasing temporal trend of β_2_-MG only in children with dehydration, suggesting early renal damage in these children. 

MA is a sign of diabetic nephropathy and a reliable diagnostic indicator for early renal damage, and it is a risk factor of cardiovascular diseases [37,38]. Given that the glomerular basement membrane (GBM) is a filtration barrier, when it is dysfunctional it can lead to the development of MA [39]. Three possible mechanisms, including glomerular endothelial dysfunction, intraglomerular hypertension and hemodynamic maladjustment, and podocyte injury, may have caused the occurrence of MA [40]. Studies have shown that increased water intake is associated with a low risk of chronic kidney disease and albuminuria [41] and early renal mild lesions are reversible [13]. Our study found a decreasing temporal trend over the school week of MA only in the children with euhydration. When β_2_-MG is elevated ahead of MA, this suggested earlier tubule impairment, while MA reflected glomerular damage. 

There was an increasing temporal trend of β_2_-MG/MA in both the euhydration and the dehydration status. Therefore, we consider that children with dehydration are at greater risk for early renal impairment [42], perhaps reflecting an accumulated burden of school time in a highly competitive academic environment. The kidney is an important organ of the body that ensures the stability of the body’s internal environment and maintains normal metabolism [37]. As a robust demonstration of renal impairment, we observed an increasing prevalence of a temporal trend for early tubular stress and a stable prevalence of glomerular damage among children with dehydration, while an increasing prevalence of tubule damage markers and a decreasing prevalence of glomerular damage markers were noted among children with euhydration. These findings indicate that, under the same weekly activity patterns, the kidneys of dehydrated children are under greater stress, and water deprivation leads to increased impaired reabsorption via renal tubules and the elimination of more microglobulin. Glomerular damage among children in a euhydration status is partially reversed with adequate water intake, which suggest a potential protective effect of optimal hydration status, confirmed by a higher but nonsignificant prevalence of overall renal impairment in children in a euhydration status. In this study, we confirm that dehydration status can aggravate early renal impairment. Thus, early renal impairment can be prevented by encouraging water intake.

The strengths of our study include the longitudinal investigation of a large sample of essentially healthy children to explore the effect of hydration status on renal impairment. Moreover, a narrow age span before puberty of the study participants enabled us to make a robust and stable conclusion for these second grade children. Another strength is a novel indicator, i.e., the ratio of β_2_-MG/MA, that we generated to evaluate the overall renal impairment. We will continue validating this indicator through the follow-ups of the current and other prospective cohort studies involving Chinese children. Our data suggesting that dehydration status aggravates early renal impairment is robust.

Our study has several limitations. First, we use the USG in morning urine to assess hydration status; this may overestimate the prevalence of dehydration. Nevertheless, studies have shown that USG and urine osmolality are strongly correlated, and USG is recommended to be used for assessing hydration status in large population studies [43]. Second, we did not include data related to children’s lifestyle and curricula patterns during the study period. We speculate that, due to the social and cultural environment, children attending primary schools in Beijing are under substantial academic pressures [9] and the opportunities for physical activity do not meet the recommended standard during the school week. Our findings underscore the necessity in encouraging more water intake for the children in the long sedentary time periods of their school weeks.

## 5. Conclusions

Our study documented different temporal trends in measures of renal impairment, comparing Monday, Wednesday, and Friday in the school week among school-age children in a euhydration or dehydration status. Children with dehydration were at greater risk of early renal impairment, especially in tubular stress/damage. To address early renal impairment in childhood, it is critical to pay attention to the hydration status and water intake for children’s health.

## Figures and Tables

**Figure 1 nutrients-14-00335-f001:**
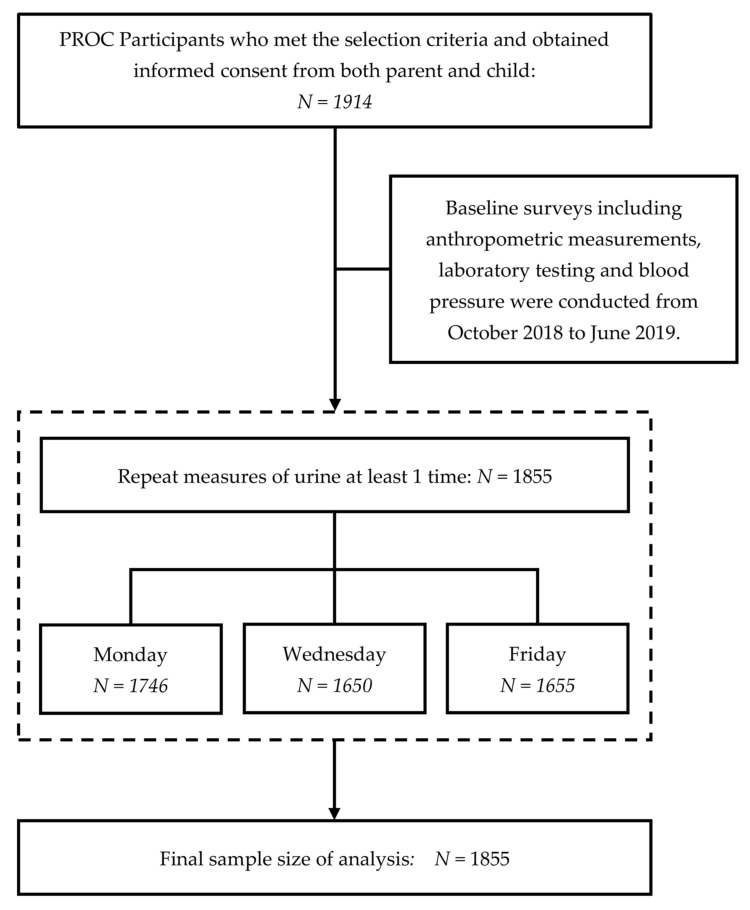
Flowchart of the participants’ recruitment and selection process.

**Figure 2 nutrients-14-00335-f002:**
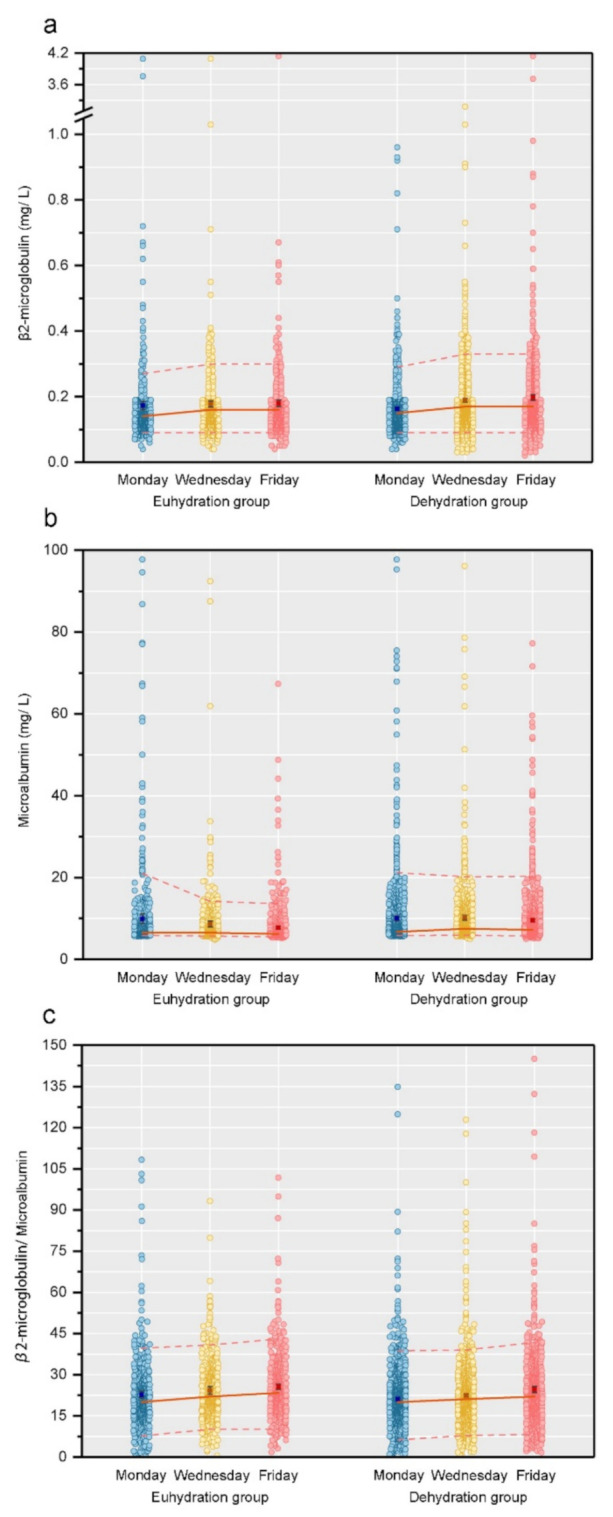
Scatter diagram of 3 urine indicators over three repeated measurements by hydration status groups among school children in Beijing. (Illustrations: (**a**) for β_2_-microglobulin; (**b**) for microalbumin; (**c**) for β_2_-MG/MA.)

**Table 1 nutrients-14-00335-t001:** Descriptive characteristics of 7–8 years old children categorized by hydration status in Beijing, China (*N* = 1855).

Factors	Total	Euhydration	Dehydration	*p*
Sex ^1^				0.033
Boy (*n* (%))	927 (50.0)	331 (46.8)	596 (51.9)	
Girl (*n* (%))	928 (50.0)	376 (53.2)	552 (48.1)	
Age (year) ^2,3^	6.7 ± 0.30	6.7 ± 0.31	6.6 ± 0.29	0.16
Z-score of height ^2^	0.67 ± 1.40	0.53 ± 1.33	0.75 ± 1.44	0.83
Z-score of weight ^2^	0.61 ± 0.95	0.60 ± 0.95	0.61 ± 0.96	<0.001
Z-score of BMI ^2^	0.40 ± 1.53	0.21 ± 1.46	0.51 ± 1.56	<0.001
WC (cm) ^2^	56.69 ± 7.81	56.05 ± 7.17	57.08 ± 8.16	0.004
SBP (mm Hg) ^2^	101 ± 8	101 ± 8	102 ± 9	0.049
DBP (mm Hg) ^2^	56 ± 6	56 ± 6	56 ± 6	0.96
BGLU (mmol/L) ^2^	5.07 ± 0.47	5.05 ± 0.44	5.08 ± 0.48	0.16
TC (mmol/L) ^2^	4.53 ± 1.07	4.52 ± 0.94	4.54 ± 1.15	0.57
TG (mmol/L) ^2^	0.69 ± 0.28	0.67 ± 0.27	0.70 ± 0.29	0.009
HDL-C (mmol/L) ^2^	1.62 ± 0.32	1.64 ± 0.30	1.61 ± 0.33	0.08
LDL-C (mmol/L) ^2^	2.50 ± 0.64	2.47 ± 0.66	2.52 ± 0.63	0.09
HbA1c (%) ^2^	5.46 ± 0.26	5.47 ± 0.25	5.45 ± 0.26	0.09
USG	1.020 (1.015–1.025)	1.015 (1.010–1.015)	1.025 (1.020–1.025)	<0.001
β_2_-MG (mg/L) ^4^				
Monday	0.14 (0.12–0.18)	0.14 (0.12–0.18)	0.15 (0.12–0.18)	0.52
Wednesday	0.16 (0.13–0.21)	0.16 (0.13–0.19)	0.17 (0.14–0.21)	<0.001
Friday	0.16 (0.13–0.21)	0.16 (0.13–0.20)	0.17 (0.13–0.22)	<0.001
MA (mg/L) ^4^				
Monday	6.60 (6.10–8.50)	6.50 (6.00–8.20)	6.70 (6.10–8.70)	<0.001
Wednesday	7.00 (6.20–9.10)	6.50 (6.00–8.00)	7.50 (6.50–9.80)	<0.001
Friday	6.70 (6.00–8.90)	6.20 (5.80–7.30)	7.20 (6.20–9.70)	<0.001

(BMI: body mass index; WC: waist circumference; SBP: systolic blood pressure; DBP: diastolic blood pressure; BGLU: blood glucose; TC: total cholesterol; TG: triglyceride; HDL-C: high-density lipoprotein cholesterol; LDL-C: low-density lipoprotein cholesterol; HbA1c: hemoglobin A1c; USG: urine specific gravity; β_2_-MG: β_2_-microglobulin; MA: microalbumin.) ^1^ Comparison of sex by hydration status using χ^2^ test. ^2^ Mean and standard deviation (SD) compared by hydration status using independent *t*-tests. ^3^ Age listed in the table is the baseline survey from October to November 2018 is the age for height, weight, WC, BMI, and blood metabolic panels, other than for repeat urine specimen collection. ^4^ Median and interquartile ranges (IQR) compared by hydration status using the Mann–Whitney U test.

**Table 2 nutrients-14-00335-t002:** Temporal changes of renal damage indicators within one week, grouped by hydration status among 7–8 years old children, Beijing, China.

Dependent Variables	Independent Variables	Model 1		Model 2		Model 3		Model 4	
		Estimate (95% CI)	*p*	Estimate (95% CI)	*p*	Estimate (95% CI)	*p*	Estimate (95% CI)	*p*
β_2_-MG									
	Euhydration								
	Intercept	0.172 (0.154,0.189)	<0.001	0.155 (0.124,0.185)	<0.001	0.156 (0.126,0.186)	<0.001	0.151 (0.120,0.182)	<0.001
	Monday	ref.		ref.		ref.		ref.	
	Wednesday	0.007 (−0.014,0.028)	0.52	0.007 (−0.014,0.028)	0.52	0.007 (−0.014,0.028)	0.52	0.007 (−0.014,0.028)	0.52
	Friday	0.005 (−0.013,0.023)	0.58	0.005 (−0.013,0.023)	0.58	0.005 (−0.013,0.023)	0.58	0.005 (−0.013,0.023)	0.58
	**Dehydration**								
	Intercept	0.163 (0.156,0.169)	<0.001	0.167 (0.148,0.185)	<0.001	0.166 (0.148,0.183)	<0.001	0.166 (0.147,0.184)	<0.001
	Monday	ref.		ref.		ref.		ref.	
	Wednesday	0.029 (0.018,0.040)	<0.001	0.029 (0.018,0.040)	<0.001	0.029 (0.018,0.040)	<0.001	0.029 (0.018,0.040)	<0.001
	Friday	0.035 (0.024,0.046)	<0.001	0.035 (0.024,0.046)	<0.001	0.035 (0.024,0.046)	<0.001	0.035 (0.024,0.046)	<0.001
**MA**									
	**Euhydration**								
	Intercept	9.502 (8.078,10.926)	<0.001	7.631 (5.469,9.794)	<0.001	7.623 (5.456,9.790)	<0.001	7.747 (5.503,9.990)	<0.001
	Monday	ref.		ref.		ref.		ref.	
	Wednesday	−0.866 (−2.564,0.832)	0.32	−0.866 (−2.564,0.832)	0.32	−0.866 (−2.564,0.832)	0.32	−0.866 (−2.564,0.832)	0.32
	Friday	−1.822 (−3.211,−0.432)	0.010	−1.822 (−3.211,−0.432)	0.010	−1.822 (−3.211,−0.432)	0.010	−1.822 (−3.211,−0.432)	0.010
	**Dehydration**								
	Intercept	10.333 (9.249,11.417)	<0.001	6.531 (4.766,8.296)	<0.001	6.468 (4.731,8.205)	<0.001	6.311 (4.513,8.110)	<0.001
	Monday	ref.		ref.		ref.		ref.	
	Wednesday	−0.258 (−1.430,0.915)	0.67	−0.258 (−1.430,0.915)	0.67	−0.258 (−1.430,0.915)	0.67	−0.258 (−1.430,0.915)	0.67
	Friday	−0.727 (−1.779,0.325)	0.18	−0.727 (−1.779,0.325)	0.18	−0.727 (−1.779,0.325)	0.18	−0.727 (−1.779,0.325)	0.18
**β_2_-MG/MA**									
	**Euhydration**								
	Intercept	22.562 (20.670,24.454)	<0.001	24.367 (20.596,28.138)	<0.001	24.367 (20.626,28.108)	<0.001	23.792 (19.896,27.687)	<0.001
	Monday	ref.		ref.		ref.		ref.	
	Wednesday	1.928 (−0.622,4.478)	0.14	1.928 (−0.622,4.478)	0.14	1.928 (−0.622,4.478)	0.14	1.928 (−0.622,4.478)	0.14
	Friday	2.963 (0.916,5.009)	0.005	2.963 (0.916,5.009)	0.005	2.963 (0.916,5.009)	0.005	2.963 (0.916,5.009)	0.005
	**Dehydration**								
	Intercept	21.008 (20.204,21.812)	<0.001	24.704 (22.161,27.247)	<0.001	24.569 (22.113,27.024)	<0.001	24.875 (22.373,27.377)	<0.001
	Monday	ref.		ref.		ref.		ref.	
	Wednesday	1.687 (0.562,2.813)	0.003	1.687 (0.562,2.813)	0.003	1.687 (0.562,2.813)	0.003	1.687 (0.562,2.813)	0.003
	Friday	3.559 (2.188,4.931)	<0.001	3.559 (2.188,4.931)	<0.001	3.559 (2.188,4.931)	<0.001	3.559 (2.188,4.931)	<0.001

(BMI: body mass index; WC: waist circumference; SBP: systolic blood pressure; DBP: diastolic blood pressure; BGLU: blood glucose; TC: total cholesterol; TG: triglyceride; HDL-C: high-density lipo-protein cholesterol; LDL-C: low-density lipoprotein cholesterol; HbA1c: glycated hemoglobin; β_2_-MG: β_2_-microglobulin; MA: microalbumin.) Model 2: adjusting sex, Z-score of BMI. Model 3: adjusting sex, standardized WC. Model 4: adjusting sex, Z-score of BMI, standardized WC, standardized SBP, standardized DBP, standardized BGLU, standardized TC, standardized TG, standardized HDL-C, standardized LDL-C, and standardized HbA1c.

**Table 3 nutrients-14-00335-t003:** Temporal trends of renal impairments by dehydration status over the school week in 7–8 years old children, Beijing, China (*N* = 1855).

Indicator	Euhydration	Dehydration	χ^2^	*p*
Elevated β_2_-MG				
Monday	111 (15.7)	188 (16.4)	0.15	0.70
Wednesday	149 (21.1)	327 (28.5)	12.59	<0.001
Friday	161 (22.8)	373 (32.5)	20.16	<0.001
Elevated MA				
Monday	37 (5.2)	62 (5.4)	0.02	0.88
Wednesday	19 (2.7)	60 (5.2)	6.92	0.001
Friday	13 (1.8)	59 (5.3)	12.78	<0.001
Elevated β_2_-MG/MA				
Monday	34 (4.8)	46 (4.0)	0.68	0.41
Wednesday	41 (5.8)	47 (4.1)	2.82	0.09
Friday	54 (7.6)	74 (6.5)	0.97	0.33

(β_2_-MG: β_2_-microglobulin; MA: microalbumin.)

## Data Availability

The data that support the findings of this study are not publicly available but are available from the corresponding author on reasonable request.

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
