# Peer review of "Dehydration Status Aggravates Early Renal Impairment in Children: A Longitudinal Study"

_nutrients, 2022, doi:10.3390/nu14020335_

Round 1
Reviewer 1 Report
There are some points raised my concern in the manuscript:
In the present study, authors try to investigate the trends of β2-MG and MA at different hydration states during a week, and to assess the effect of hydration state on early childhood renal function. However , there are some points raised my concern:
- Is it possible to understand the difference of the fluid status during school days and weekends of the study population ? Since authors try to investigate the different hydration states " during a week"?
- Why the urine samples only collected on Monday , Wednesday and Friday, and not everyday ? Is there any reasons for that? Please clarify.
- I suggest the author to present the sex in either boy or girl instead of both to simplify the presentation.
Author Response
- Is it possible to understand the difference of the fluid status during school days and weekends of the study population? Since authors try to investigate the different hydration states " during a week"?
Response:Thank you for the insightful comments and sorry for the confusion. We would like to explore the temporal trends of renal impairment among children in different hydration states from Monday to Friday. The weekly activity pattern is an accumulated process of the sedentary time during school time. While it is a recovering process of renal damage during weekend break given the unpublished data of time activity patterns of PROC participants, a significant difference in physical activity time between weekdays (average18 minutes, daily) and weekend (72 minutes, daily). We conducted repeated measures of urinalysis in three weekdays (Monday, Wednesday and Friday): Monday reflected renal status over a break, Wednesday reflected intermediate time of the sedentary time accumulation , and Friday reflected the accumulation of school time for almost one week .
- Why the urine samples only collected on Monday, Wednesday and Friday, and not everyday? Is there any reasons for that? Please clarify.
Response: Thank you for your query. As we responded in the first question, the three days were chosen to explore the variations of different weeks of the workday. Moreover, to save cost, we did not collect and assay for everyday of the week. And we have reworded the contents of “2.1 Study design and participants” section. And paragraphs 2 now reads as: “We conducted three repeat measures of urinalysis (UA) in two weeks of November 2019. The urine samples were consecutively collected in the morning of three weekdays (Monday, Wednesday and Friday) within one week to comprehensively explore the re-nal damage trends during school time each week: Monday reflected renal status before the start of the school week after fully “resting” over the weekend, Wednesday reflected intermediate time, and Friday reflected the accumulation of almost the whole week at school.”
- I suggest the author to present the sex in either boy or girl instead of both to simplify the presentation.
Response: Thank you for the suggestion. We have revised the expression and kept the results for boys for clarification in line 21, p1, line159, p5, line239, p10, and line244, p10. Wish it make sense.
Reviewer 2 Report
Scientific contribution
The article, as declared by authors, is the first longitudinal study to assess the relationship between hydration status and renal function in children living their daily routine, including 1914 children enrolled in a community-based census-like designed cohort for the PROC study conducted in six public non-boarding primary schools in Shunyi District in Beijing.
Materials and methods
Three urine markers were considered to assess hydration status and renal function impairment.
Urine specific gravity was used to assess hydration status.
Measurements were not performed using the same analyzer, but due to the large number of samples there were used 3 different instruments. It’s not clear whether measurements were performed by the same laboratory. Furthermore, the marker seems to be inappropriate 1,2,3 to assess hydration status.
Urinary beta-2 microglobuline and microalbuminuria were used to assess renal function impairment, but urinary B2M and albumin are markers of renal damage. Furthermore, there is no universally accepted relationship between urinary b2m and risk of CKD onset and/or progression.
Discussion
Dehydration prevalence was comparable to other studies cited by authors in which dehydration was assessed by urine osmolality, which is considered a better performing marker compared to urine specific gravity.
Results showed higher mean urine specific gravity between male sex and females. There’s a difference between mean square z-score in “euhydrated” and “dehydrated” group, thus suggesting that BMI and sex could be a risk factor for dehydration. Further evidence are needed.
Role of urinary beta-2 microglobuline and macroalbuminuria as markers of proximal tubule and endothelial disfunction are poorly referred.
According with authors, the strength of this study is the longitudinal investigation of a large sample of essentially healthy children living their daily routine, thus investigating the impact of social, cultural and economic factors on the adequacy of fluid intake in children, given a sufficient water intake as a nefroprotective factor.
Referee’s conclusion
A major revision is needed since markers chosen by the authors are poorly performing in assessing hydration status and markers chosen to assess kidney function impairment seem to be more appropriate to define a kidney injure. Furthermore, stronger references are needed.
Novel marker B2-MG/MA could need a validation in an external cohort set.
Author Response
1.It’s not clear whether measurements were performed by the same laboratory.
Response: Thank you for the comments and sorry for the confusion. Urine specific gravity (USG) was measured by two laboratories and three instruments. We reworded the contents of “2.6. Hydration Status” and the text now reads: “USG was measured by two laboratories ( both of them were audited and certified by the National Medical Products Administration) and three instruments: one laboratory used URIT-500B (Guangxi, China), another used Dirui H-800 (Guangdong, China) and Combi Scan 500 (Lichtenfels, Germany)”.
2.Furthermore, the marker seems to be inappropriate 1,2,3 to assess hydration status.
Response: Thank you for your insightful comment. We have acknowledged this as limitation. Although defining hydration status by urine osmolality is a more reliable way, it is difficult to collect 24h urine for multiple times in large-sample study and we collected once, the specimens on Mondays that were 24h urine collection. Studies showed that urine specific gravity has similar specificity (91%) and sensitivity (89%) compared to urine osmolality, and it has the advantage of low inter-individual variability. And also using urine specific gravity to define hydration status can reduce clinical and economic burden. We don’t understand what “1,2,3” means, so please forgive us if we do not explain the content.
3.Urinary beta-2 microglobuline and microalbuminuria were used to assess renal function impairment, but urinary B2M and albumin are markers of renal damage. Furthermore, there is no universally accepted relationship between urinary b2m and risk of CKD onset and/or progression.
Response: Thank you for your constructive comments. According to your suggestions, we have revised the description and updated the corresponding references. Though the relationship between urinary β2-MG and risk of CKD onset was not fully determined, urinary β2-MG has been investigated as candidate biomarker for acute kidney injury and diagnosing renal diseases in child with variable success. We sincerely hope this article could enrich current research and report a vigilant predictor for a preventable renal damage from harmful life style in children, like long sedentary time in school time.
Discussion
Dehydration prevalence was comparable to other studies cited by authors in which dehydration was assessed by urine osmolality, which is considered a better performing marker compared to urine specific gravity.
4.Results showed higher mean urine specific gravity between male sex and females. There’s a difference between mean square z-score in “euhydrated” and “dehydrated” group, thus suggesting that BMI and sex could be a risk factor for dehydration. Further evidence are needed.
Response: Thank you for your suggestions. We have reworded the contents of discussion section. And the third paragraph of discussion now read as: “We observed differences in the Z-score of body weight, Z-score of BMI, and WC between the euhydrated and the dehydrated group. This finding validates a previous study that children with excess body fat have a higher risk of dehydration[31]. Children with dehydration have higher BMI Z-score and have higher water requirements, because water requirements related to higher metabolic rate, body surface area and body weight. Furthermore, obese children are more likely to consume foods which are higher in calories, yet lower in water [32]. Moreover, we observed that children with dehydration were more likely to have higher SBP, TG, and LDL-C and lower HbA1c (the latter two were of borderline significance) because dehydration may increase angiotensin II and blood viscosity and may impair endothelial function [33]”
5.Role of urinary beta-2 microglobuline and macroalbuminuria as markers of proximal tubule and endothelial disfunction are poorly referred.
Response: Thank you for your constructive comments. We expanded the content in the discussion section in line 284-306, p11. It now reads: “We found that children with dehydration status had higher β2-MG and MA than those in euhydration status. β2-MG is a low molecular weight protein which is filtered by the glomerulus and completely reabsorbed and catabolized in the renal tubules[34]. Under normal physiological conditions, only a small amount of β2-MG can be detected in the urine and the elevated concentration of urine imply proximal tubular dysfunction[35]. Among children, tubular dysfunction can be used to diagnose tubulo-interstitial diseases, localization of UTI. Furthermore, β2-MG has been reported as candidate biomarker for acute kidney injury (AKI)[36]. We found an increasing temporal trend of β2-MG only in children with dehydration, suggesting early renal damage in these children. MA is a sign of diabetic nephropathy and a reliable diagnostic indicator for early renal damage, and it is a risk factors of cardiovascular diseases[37, 38]. Given that the glomerular basement membrane (GBM) is a filtration barrier, when it is dysfunctional it can lead to the development of MA[39]. Three possible mechanisms including glomerular endothelial dysfunction, intraglomerular hypertension and hemodynamic maladjustment, and podocyte injury may have caused the occurrence of MA[40]. Studies have shown that increased water intake is associated with a low risk of chronic kidney disease and albuminuria[41] and early renal mild lesions are reversible[13]. Our study found a decreasing temporal trend over the school week of MA only in children with euhydration. When β2-MG is elevated ahead of MA, this suggests earlier tubule impairment, while MA reflects glomerular damage. There was an increasing temporal trend of β2-MG/MA in both euhydration and dehydration status. Therefore, we consider that children with dehydration are at greater risk for early renal impairment[42], perhaps reflecting an accumulated burden of school time in a highly competitive academic environment”.
6.According with authors, the strength of this study is the longitudinal investigation of a large sample of essentially healthy children living their daily routine, thus investigating the impact of social, cultural and economic factors on the adequacy of fluid intake in children, given a sufficient water intake as a nefroprotective factor.
Response: Yes, Thank you for the summative highlights.
Referee’s conclusion
- A major revision is needed since markers chosen by the authors are poorly performing in assessing hydration status and markers chosen to assess kidney function impairment seem to be more appropriate to define a kidney injure. Furthermore, stronger references are needed.
Response: Yes, Thank you and we strengthen the discussion accordingly in addressing the item 5th comments.
8.Novel marker B2-MG/MA could need a validation in an external cohort set.
Response: Thank you for the constructive comments and we acknowledge this in the discussion line 308, p12, “We will continue validating this indicator through the follow-ups of the current and other prospective cohort studies involving Chinese children.”
Round 2
Reviewer 2 Report
All issues raise has been adequately addressed; still remaining some intrinseca problem related to study design and sample collection, I think that it can be useful to highlight the attention of scientific community on a simple but very important point.